

# The multiple understandings of wind turbine noise: Reviewing scientific attempts at handling uncertainty.

Julia K. Kirkegaard[1], Tom H. Cronin[1], Sophie Nyborg[1], Daniel N. Franzten[1]

[1]DTU Wind and Energy Systems, Technical University of Denmark, Roskilde, 4000. Denmark

*Correspondence to:* Julia K. Kirkegaard (jukk@dtu.dk)

**Abstract.** The noise from wind turbines has been an issue in the planning and development of wind power for many years, giving rise to both controversies during the deployment of onshore wind farms as well as a significant amount of research by various communities of scientists, or what we treat here as epistemic communities. Despite iterative attempts at fixing the

noise issue through investments into technological developments and regulatory determination of allowable decibel noise levels, noise remains a contested and difficult object to find solutions to. In the Co-Green project, we instigated a social science-based study founded in Science & Technology Studies (STS) to look at why and how it is that noise continues to be so controversial. We do this through a narrative literature review of three different epistemic communities – the technical, health-based, and social acceptance literatures – tracing the emergence of the knowledge object of wind turbine noise. We

illustrate how noise remains an 'unruly knowledge object' that defies stabilisation within and between the three epistemic communities: Instead, noise is understood as fundamentally different things across the three communities, fuelling the controversies over the solutions proposed, where the "fixes" might sometimes not address what was intended. We end by pointing to the potential benefits of more interdisciplinary engagement between epistemic communities and as well as to - in the context of science for policy - probe the potential value of finding ways to translate qualitative research findings into

noise (and other) regulations.

## 1 Introduction

The deployment of wind farms is considered by many countries as an important activity in order to meet targets to reduce the levels of greenhouse gas emissions by the electricity sector. However, with onshore wind power still being the cheapest option, the continued efforts to install wind turbines on land are meeting sustained opposition from local communities,

leading to stalled and even cancelled wind power projects (Ellis and Ferraro, 2016). This highlights the need for a better understanding of the 'social grand challenge' of wind energy (Kirkegaard et al., 2023).

One of the most contested issues in wind farm developments has been the "sound" emanating from wind turbines (Borch et al., 2020; Solman et al., 2023; Wind2050.dk), constituting it as problematic or even politicized "noise". The issue of wind



turbine noise is a mandatory topic in the Environmental Impact Assessment (EIA) for the planning of a wind farm and is the subject of regulations that, in many countries, are made specifically to apply to the sound from wind turbines. However, despite many of the stakeholders involved in the deployment of wind farms being bound by these regulations (e.g. municipal planners, developers, wind turbine manufacturers, and environmental consultants) there continues to be many disputes around wind turbine sound. Common issues raised are concerned with how the regulations are set, the legitimacy of the

noise levels that are allowed, their calculation, measurement and certification according to IEC standards, and the potential health impacts of the noise on citizens living around wind farms.

These issues and their implications for wind farm development are typical of those seen in Denmark, and there has been significant funding of ambitious projects to resolve the issue. One example is the study commissioned by the Danish

government and conducted by Denmark's Cancer Society (Poulsen et al., 2019), which involved a nation-wide analysis of the health conditions of people living in the vicinity of wind turbines, correlated with calculations of the noise levels they would experience. Another example is the construction of the Poul la Cour wind tunnel (Plct.dk) at the Technical University of Denmark's (DTU) Wind and Energy Systems Department at Risø campus, which is reputed to be one of the biggest university-owned wind tunnels in the world (Videnskab.dk), designed specifically for measurements of wind turbine

aerodynamics and noise. A third example is the organization "Viden om Vind" ("Knowledge about Wind"), supported by various wind farm developers in Denmark, which has as its stated aim to provide more facts about wind energy, to inform local communities about the issue of noise, amongst other things, and to form a basis for informed dialogue amongst citizens, public authorities, and politicians (videnomvind.dk).

Central to these various undertakings is the idea of establishing facts about noise levels, universally calculated and measured using the unit of decibels (dB). In the Co-Green project on "Controversies in the green transition: The case of wind turbine sound and its politicisation" (see dff.dk; Independent Research Fund Denmark 2021-2024), we hypothesise that isolating noise like this through a "one-dimensional" techno-scientific metric of dB, means that the standard response to the challenges of noise and wind power in the green transition has primarily been the implementation of technological solutions

to reduce the dB level. Yet, this comes, we argue, with the risk of disregarding important non-technical, and less quantifiable, concerns and forms of knowledge that may lie at the root of social controversies about wind farm developments. In this paper, we aim to find out how noise is understood by different scientific literatures and the effects these different understandings have on the solutions to the issue of noise that they propose.

We start with a critical and historical review of the literature on wind turbine noise, where we identified three key literatures, which we have labelled as Technical, Health, and Social Acceptance. We then illustrate how different understandings of wind turbine noise that are produced by scientific communities are not 'neutral' but have material effects on what noise is perceived as, and ultimately how noise issues are addressed. This finding leads us to call for reflexivity on how scientific





concepts such as noise are co-produced by the scientific communities that form around them, in order to not end up with
solutions that may not actually address the issue they were intended to solve.

With this, our analysis points to three provocative, but hopefully constructive, findings: 1) Noise is understood as a very
different object in the three literatures that we have reviewed. 2) Noise can be construed as a "scientific object" that defies
being controlled or "pacified" and is unruly because it is not just a technical issue but also a socio-technical one. 3) Third,
while the technical and health-related studies believe that they are addressing the issue of social acceptance with their work,
they are in effect dealing with what they understand is a proxy for social acceptance. We end by discussing how these
findings might be understood in the context of how scientific communities try to deal with uncertainty surrounding scientific
objects such as noise.

Finally, we would like to highlight that this study is based on social science techniques and methods, and a secondary "meta-
level" purpose of this paper is deliberately to try to make such a study relevant for a technical audience such as that of the
WES (Wind Energy Science) readership. Our approach is to try to be challenging but helpful, and to highlight different
perspectives to promote a reflexive response in the reader, ultimately hinting at the challenges of interdisciplinary
considerations and the prospect of one's own research being influential far beyond what is conventionally expected (also see
Kirkegaard et al., 2023, Nyborg et al., forthcoming).

## 2 Conceptual framework

In a social science study, theory and methodology have a somewhat different position than in the technical domain. To
measure sound power levels with a thermometer would be unthinkable to an engineer, but to social science scholars, their
theoretical lens(es) may be used to look at both music and temperature, so to speak. It is thus essential to state the theoretical
basis on which one is building, the relevant scholarly works, and how they are being used, as this fundamentally informs the
way in which the study is to be understood. Here, we give a short summary of the conceptual and theoretical foundation for
this paper.

In this study, we build upon the literature within Science and Technology Studies (STS) about how scientific facts and
objects are made (e.g. Latour, 1980, 1987; Callon, 1980; Knorr-Cetina, 1997, 1999; Collins and Evans, 2007). We do so by
relying particularly on Karin Knorr-Cetina's (1997, 1999) notions of "epistemic communities" and "epistemic
objects"/"knowledge objects". In our approach, we consider epistemic communities to consist of specific literatures and
experts, and their various tools and instruments, inquiring into how different epistemic communities produce particular
understandings and "facts" around emergent knowledge objects as well as groups of problems and solutions around the issue



of noise, as already investigated with regard to noise (see e.g. Bijsterveld 2001; Pinch and Bijsterweld, 2004; Taylor and Klenk, 2019).

From our theoretical standpoint, epistemic communities frame the issue of wind turbine noise in particular ways. This framing process often entails acts of what we call purification, isolating it from other issues through compartmentalisation
and disentanglements. Meanwhile, stabilisation of a knowledge object through purification and disentanglement – constituting it as a stabilised fact – may not always be feasible, particularly if the knowledge object produces conflicting data about itself or if purification simplifies so much so that it overlooks some of the entanglements, causing new unforeseen issues to emerge. Indeed, knowledge objects tend to remain incomplete, constantly mutating, defined as much by what they are not as by what they are and often "exist[ing] simultaneously in a variety of forms" (Knorr-Cetina, 1999, pp. 14-15).

In this paper, we investigate different attempts at stabilising the epistemic object of wind turbine noise by different epistemic communities, revealing that it remains an unruly epistemic object that refuses to be fixed and entirely stabilised. At the same time, we examine how the different understandings also have material impact on how the issue of noise is framed as a problem, and the types of solutions that are proposed.

## 3 Methodology: Mapping the evolution of the noise issue

In order to gain a rich understanding of how the epistemic object of wind turbine noise has evolved over time, we combine our literature review with expert interviews and ethnographic field observations. That is, while the literature review attuned us to how the epistemic object of wind turbine noise has emerged historically, our emergent interpretations were cross-checked and triangulated by experts in the field.

First and foremost, our study is based on an extensive narrative literature review (Greenhalgh et al., 2005), using scientific databases (Scopus, Web of Science, and Pub-med) to search for the various storylines of tasks, problems and solutions concerning wind turbine noise, using search strings developed for each literature (see appendix A for examples of search-strings). Based on this search, we identified three main literatures (or 'epistemic communities') that construe wind turbine
noise in particular ways: 1) "Technical" (engineering and acoustics), 2) "Health" (psychology and medicine), and 3) "Social Acceptance" (social science). We acknowledge that our analysis presents these three epistemic communities as somewhat separate "ideal types", but we also recognise that there are overlapping interests and engagement. For instance, we have identified attempts at establishing cross-disciplinary collaborations between the technical and health-based communities, and sometimes the literatures refer to each other's work, using it as legitimisation for their own research (Nyborg et al.,
forthcoming).



Based on our literature reading, we categorised the different understandings of noise by tracing the following aspects. 1) Historical background (the study of noise); 2) How noise is understood (how is it "seen? What is it? How is it disentangled?); 3) How noise is found (and with what tools, techniques, instruments?), and 4) How it is being treated or solved (what tools, techniques, instruments?). We supported our findings from the review with a total of 12 recorded and transcribed interviews with representatives from the three epistemic communities plus an expert in noise regulations and standards. Interviewees were found through the snowballing method. (See appendix B for a list of anonymized interviews.) During the interviews, we used a visual timeline of events that was customized to each interview situation. This allowed us to corroborate our understanding of milestone events in the historical development of noise as a field of study. (See example of timeline used in interviews, Appendix C). A third source of data came from participant observations at the Wind Turbine Noise conference in 2021 (online, INCE 2021) and participation in person at the 2023 version of this conference in Dublin, Ireland (INCE 2023). Further observations were made at project meetings of the International Energy Agency's (IEA) Task 39 on 'Quiet Wind Turbine Technology' and during a noise measurement campaign in Jutland, Denmark, conducted by DTU engineers (iea.wind.org.). These interviews and observations helped to inform, qualify and test our analysis that had resulted from the literature review, and we use a few direct quotes from interviews and observations in this paper.

## 4 Attempts at taming the unruly object of wind turbine noise

Our analysis maps out how the three epistemic communities (Technical, Health and Social Acceptance) have understood the subject of noise, and how they have attempted to 'tame' it. We do this by looking at four aspects, namely 1) the historical background of each community's research, which has led to 2) an understanding of how they view and understand noise and try to isolate it in their studies. We have also studied 3) what manner of tools and techniques they use to identify noise, and finally 4) how they formulate the issue, treat it and, if relevant, how they try to solve the noise problem. We thus get a picture of how each epistemic community attempts to make sense of a phenomenon that evidently has attracted attention not only from researchers and practitioners in the field, but also from policymakers and local communities, and beyond.

### 4.1 Noise in the Technical community

#### 4.1.1 How is sound understood?

In our analysis, the technical epistemic community encompasses engineering, acoustics and natural sciences, and primarily works with the design, manufacturing, installation or operation of wind turbines. As an object of inquiry, the study of noise involves examination of the generation and propagation of the sound itself. Sound and noise are measured in the same acoustic unit, that is, the decibel (or dB), that was originally coined for audio levels in telephone cables back in the early 1900s (Garret 2020, p. 466 on the Bell Labs). As the human ear is not equally sensitive to all frequencies (first measured by Fletcher and Munson, 1933), then a method of averaging (or weighting scale) has been developed and is now governed by



the international standard ISO 226 (International Standardization Organization). While the so-called A-weighting (dB(A)) inevitably masks the individual frequencies of the original sound, it is by far the most commonly used metric.

### 4.1.2 From sound to noise

When propagating and traveling to our ears, sound vibrations cause the vibration of the human ear drum, which in turn are translated into impulses that the brain perceives as sound (Gunther, 2012, p. 306). Pure sound (e.g. a musical note where air particles vibrate in a neat/regular and predictable, and thus calculable, fashion) is hereby a very tangible and physical phenomenon. The vibrations are always depicted graphically by waves, much the same as electricity, and something that can relatively easily be shown to "obey" the natural laws of wave physics such as superposition (Gunther, 2012, p. 205).


It is when sound vibrations become uncontrollable and unpredictable, that sound is referred to not as musical but as noise in the technical epistemic community: "Although noises are sometimes not entirely unmusical, and notes are usually not quite free from noise, there is no difficulty in recognising which of the two is the simpler phenomenon. There is a certain smoothness and continuity about the musical note" (Rayleigh, 1945, p. 4). In contrast to pure sound, noise is more difficult

to simulate/calculate: while noise still obeying the laws of physics, it is difficult to demonstrate through equations/calculations that it does so, as it "does not so easily yield to conventional mathematical analysis" (Garret, 2020, p. 26). In general, noise is thus considered a sound that is unwanted, not useful or indicating that something is wrong, and the physics of air particles reflects this (Lee and White, 1998).

### 4.1.3 Noise from wind turbines: reducing noise levels while maximising production

The noise emanating from a turbine is mostly aerodynamic noise, that is, air flowing over the blades resulting in turbulence and rapid vibration of air particles (Wagner, 1996). The design of a wind turbine blade has historically concentrated on optimising power production and is key to this epistemic community, but these blade design considerations are entangled with questions around noise. The principal fundamentals of air flowing over the blades of a wind turbine go back to the 1920s with research in the aviation industry because of the similar requirements of an aircraft wing needing the force of lift to keep the aircraft flying up in the air, and the need for the force of torque to turn the rotor of a wind turbine. So, when wind

turbine blade design development started in the 1970s, the research topic of aerofoil design was already established (Bak, 2021). Particularly important reference work was carried out by NASA in the 1950s who produced a catalogue of aerofoil designs, each tested in their wind tunnel (Abbott and von Doenhoff, 1959). The reason for the aircraft designers and wind turbine designers being interested in aerofoil design are similar: efficiency of the air flowing over the aerofoil. The more

efficient an aircraft wing is, the more lift force is generated to keep it in the air, and the more efficient a wind turbine blade is, the more torque is generated to turn the rotor, the more energy can be extracted from the kinetic energy of the wind and converted into electrical energy. This is the energy that is then sold to produce revenue for the wind turbine owner.





However, whilst noise from aircraft is mostly from the engines, noise from wind turbines is an important constraint in wind turbine blade design. Much of the engineering driver for working on wind turbine noise and its reduction is the belief that the public's perspective on sound is 'annoying noise' (Deshmukh et al., 2018). Work by Pedersen and Waye (2004) is often quoted relating perception and annoyance due to wind turbine noise (see Health section) and that a reduction of the noise level (the dose) at the human ear is a desirable goal as "noise is one of the major hindrances in the development of wind power industry" (Dai et al., 2015) as people "experience annoyance" when they live in the vicinity of wind turbines. Indeed, research by acousticians using listening tests and "idealised wind turbine sounds" (von Hünerbein, 2010) have tried to establish "audibility thresholds and equal annoyance contours" in order to quantify an impact.

However, a reduction in noise generation often comes at a price: a reduction in power produced (Wagner, 1996, p. 164). Thus, work on reducing noise must always be balanced with minimising any reduction in performance, so the energy production and the revenue from selling it, is maintained. The strong belief of the Technical epistemic community in the ability of finding a technical solution to reach such an optimum – fixing the noise issue – is reflected in the following quote: "In order to maximize energy output while complying with noise regulations, wind turbine manufacturers will continue to implement new noise reduction technologies in their products. In this way, the cost of wind power can be reduced while addressing societal concerns" (Oerlemans, 2021).

### 4.1.4 Noise generation, propagation and influence on turbine design and wind farm planning

Ultimately, the work by this epistemic community centres around predicting and measuring the noise level in dB(A) at a certain point on or away from the turbine. There are fundamentally two main areas of focus, the first concentrating on how and how much noise is generated, and the second how it propagates through the atmosphere and is received (Wagner, 1996, p. 10).

The aerodynamics of the blades has been a consistent object of research for noise generation research. When the airflow passes over the wind turbine blade "at the trailing edge", it becomes turbulent (unruly), producing a scattering mechanism which makes up the main part of aerofoil noise (Wagner, 1996, p. 67). In terms of the shape of the aerofoil, various "fixes" or adaptations have been applied, such as trailing edge serrations (or "dinotails") in order to reduce the noise from the back edge of turbine blades (Fuglsang and Oerlemans, 2012). With these devices, typically, noise reductions of about 2-3 decibel (dB) in overall A-weighted sound levels have been achieved, even if the exact mechanism is not yet understood and is still today the subject of wind tunnel testing (e.g. Ryi et al., 2014).

Another aspect of noise reduction has centred around the turbine's control system where designers can adjust the speed of rotation, together with the angle (pitch) of the blades so as to control the noise emission of a turbine. "The objective for the





low-noise controller is therefore to design optimized RPM [rotations per minute] and pitch curves, which maximize the turbine power at each wind speed within the constraints for noise, torque, etc." (Oerlemans, 2021, p.9).

Noise propagation research has focussed on being better able to predict the volume of the noise at a particular location after it has propagated in the far field (Wagner, 1996, p. 125). Yet, as expressed in one of our interviews, it remains a technical challenge to develop models that can "predict in a far field, and I mean it's a big subject for, to predict noise in a long distance because you are looking at noise levels which are very close to the background noise" (Appendix B, interview 1). When measuring the noise propagating from a wind turbine there is complex acoustical work needed to isolate the noise from the background noise (e.g. trees and vegetation, traffic, birds, etc). Even though these measurement methods are

prescribed by international standards (IEC 61400-11, 2016) measurement campaigns in the field is a cumbersome task, dependent on weather conditions, background conditions and subsequent treatment of the data to make it useful (Wagner, 1996, p.152).

The research on noise generation is mainly used in the design of wind turbines, involving disciplines from acoustical

engineering to aerodynamic and structural as well as control system engineering. On the other hand, the results from models developed by far field propagation research are often used to create noise contour maps (see Fig. 1). These maps show lines of predicted equal noise levels, often featuring the regulatory noise limits that are applicable in an area around a prospective wind farm. These very visual means of representing noise are a key feature in environmental impact assessments (EIAs) carried out during the planning of wind farm projects.


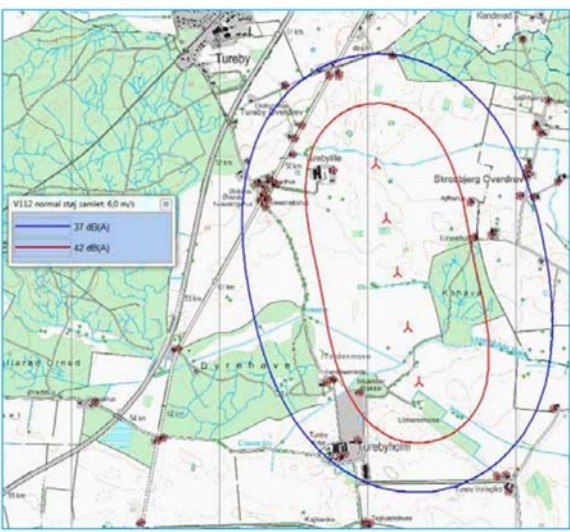

**Figure 1: Example of a noise contour map (from Turebylille wind farm EIA, 2013)**



## 4.2 Noise in the Health community

The health-based epistemic community is based upon the premise that adverse levels or types of sound can have an impact on people's health and wellbeing. This field is characterized by a variety of scientific disciplines, ranging from acoustics, natural sciences, environmental medicine, and environmental psychology.

### 4.2.1 The rise of environmental noise and the emergence of (noise) annoyance as a risk

The study of the relationship between infrastructure developments and health can be traced back to around the beginning of
the 20th century where noise figured as one amongst many environmental hazards or risks. In a study from 1910, the impacts of noise on the work efficiency of industrial workers were observed for the first time (Wynne, 1930). Since the 1960s, 'risk' studies (Kasperson et al., 1988), constituting noise in the environment as a health concern (and risk), has become established as a field in itself (foreword, ICBEN, 1973). Importantly, health is defined not just as a matter of absence of disease, but a matter of complete physical, mental, and social well-being (WHO, 1946). Therefore, the World Health Organisation (WHO)
considers (noise) annoyance as a health issue resulting from noise exposure, along with e.g. sleep disturbance, mental health, diabetes and heart disease (WHO, 2018). Annoyance is one of the most prevalent health effects of noise (Guski et al., 2017), but is moreover hypothesised to mediate a range of other 'physical' health effects (e.g. stress, anxiety, sleep disturbance, metabolic and cardiovascular disease) (WHO, 2018; Van Kamp and Van den Berg, 2018, 2021, Michaud, 2016, Taylor and Klenk, 2018). While traffic, industry and 'neighbourhood noise', for instance, have been investigated since at least the 1950s
(Stevens et al., 1955), the health impact of wind turbine noise has only been an issue in the last decades (WHO, 2018, Pedersen and Waye, 2004). Annoyance, though, continuously comes forward as the most important health consequence of wind turbine sound (Van Kamp and Van den Berg, 2021).

Low frequency and infra sound from wind turbines is also a topic of public and political concern, with concerns that these
lower, inaudible sound frequencies may lead to other health effects than audible sounds, such as the "Wind Turbine Syndrome" caused by Vibroacoustic disease (VAD) (vibration of the body, nausea or dizziness). However, there is no evidence that these lower frequencies have any health effects, nor that they lead to extra annoyance (Van Kamp and Van den Berg, 2021).

This epistemic community focuses on self-reported health effects (e.g. annoyance, stress or sleep disruption) in survey studies as well as in laboratory experiments (e.g. Müller et al., 2023; Hübner et al., 2019; Boorsma and Schepers, 2017), which also include objective measurement of bodily reactions, for instance polysomnography to detect sleep disturbance. Epidemiological studies on the population include self-reported health effects in surveys, but a couple of studies also use more objective data. The Canadian Community Noise and Health Study by Michaud et al. (e.g. 2016), for instance, used hair
cortisol concentrations to complement self-reported data. The Danish Health Study on the health-impacts of wind turbine





noise (Poulsen et al., 2019) used wind turbine location data and modelled noise levels together with medical register data in order to study the relationship between wind turbine noise and a range of medical conditions on the population.

### 4.2.2 The visibility of the dose-response graph and impact on planning

The method of modelling the relationship between a sound dose (in dB(A)) and health response, and basing research on
standardised methods to allow meta-analysis and modelling, is pervasive in the health epistemic community. The modelling of dose-response relations and predicting the health impact have a long history and have proved a key informant for science-based policy advice and for underpinning regulation.

A formative study in the 1950's by Stevens et al. (1955, p. 64) developed the "Composite Noise Rating" approach whereby
people's reaction to noise could be predicted, so that a "realistic criterion" or limit for neighbourhood noise. This was based on the recognition that levels, or doses, of noise - measured in decibels (dB(A)) - could elicit a certain human response way below that which can damage hearing, namely influencing the wellbeing and health of the general public. However, annoyance studies today build on a seminal paper on noise annoyance published in 1978 by Theodor Schultz (Schultz, 1978). He developed a model which rested on the conviction that both the noise dose and the response can be standardised
and quantified and plotted in a graph that shows visually the particular relation between sound dose and response ("% highly annoyed") for different noise sources.

The dose-response model emerged from a need to be useful for planning, regulation and policy, the raison d'être for the health-based epistemic community: In 1978, the hitherto limited agency of science to produce 'facts' for policy and planning
was problematized by professor Paul Borsky (Columbia University). In a paper presented at an environmental noise conference, Borsky argued that the "lack of agreement on standardized units of measurement and comparable methods for obtaining and analyzing objective data" in the scientific community was one of the main reasons "why community noise abatement has made such slow progress" (Borsky, 1978, p. 453).

To overcome this seeming lack of influence on policy, Schultz's overarching aim with his dose-response paradigm, which was supported in part by the U.S. Department of Housing and Urban Development (Schultz, 1978, p. 403), was to find a stable model that could better inform policymaking on land use and planning and offer "guidance for regulatory decisions about noise" (1978, p. 403; also see Fidell, 2003; Miedema and Vos, 1998, p. 3434). To do so, Schultz changed the main metric from median response (resembling an "average response" of a community) which was the common approach at the
time, to "percent highly annoyed" plotted against noise exposure. The argument was that this approach would provide a more stable (universal) model, since "the median response is much more difficult to translate from one annoyance scale to another, in everyday terms that are understood by politicians and policy makers" (Schultz, 1978, p. 379). Moreover, if the person was highly annoyed, he hypothesized the response would be strongly correlated to the sound itself, providing a more



"useful indication of acceptable community noise exposure" (Schultz, 1978, p. 379). Further, without providing further
evidence, Schultz (1978, p. 389) commented that restricting the percentage of the population being 'highly annoyed' to 10%
would be a "desirable condition". Noise limits should in other words meet that target. Pedersen and Waye (2004) conducted
the first – and now seminal – annoyance dose-response study for wind energy in 2004 to explore "acceptable exposure
levels".

"There is clearly a need for field studies to investigate the impact of wind turbines on people living in their vicinity and to
further explore the presence of disturbances. In particular, dose–response relationships should be investigated to achieve a
more precise knowledge of acceptable exposure levels" (Pedersen and Waye, 2004, p. 3461).

They concluded that wind turbine noise is a particularly annoying noise source as, despite lower doses, it produced higher
annoyance rates than other noise sources at the same dose level. (See Fig. 2). The idea that different types of noise sources
(aircraft, road traffic, train, etc.) would result in varying degrees of annoyance at similar noise levels was already put forward
by Miedema and Vos (1998). Such comparative studies on different noise sources have helped to demonstrate that wind
turbine noise may have particularly annoying characteristics (e.g. the "swishing" or "thumping" sound character, technically
referred to as amplitude modulation) that can be perceived at "low dose levels", but which are not often captured by the
measuring methodology in regulation (Janssen et al. 2011, Haggett, 2012).

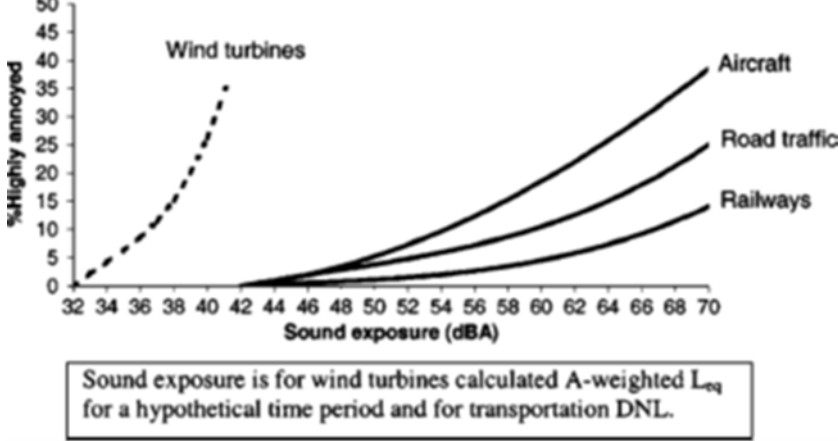

**Figure 2: Pedersen and Waye's dose-response model (2004, p. 3468)**

A 10% highly annoyed limit for wind turbine noise that Schultz argued for  figures today in the recent WHO Noise
Guidelines for the European Region (2018), which aim to "provide robust public health advice underpinned by evidence …
which is essential to drive policy action that will protect communities from the adverse effects of noise" and to "provide
policy guidance to Member States that is compatible with the noise indicators used in the EU's Environmental Noise





Directive" (2018, VII). Their conditional recommendation of a noise limit for turbines of 45 dB(A) (day-evening-night-weighted) meets the 10% highly annoyed limit.

The dose-response graphs and the scientific legitimacy it visualises, furthermore emerges in the regimes of national planning and regulation. For instance, in the Danish case, where the impact of the noise limit of 44 dB(A) is explained on the Danish Environmental Agency website as being "expected to be experienced as highly annoying for 11% of those who is subjected to it" (mst.dk).

### 4.2.3 Noise (annoyance) as an unruly epistemic object

Despite the development of comparable metrics to produce facts on health effects from wind turbine noise, annoyance continue to constitute a "conundrum" or "unruly epistemic object". As had been noted repeatedly almost since the beginning of the study of environmental noise, the sound level itself is often not the main cause of the annoyance reported. In other words, the "annoyance from a sound is not inextricably bound up with that sound' (Van Kemp and Van den Berg, 2018, p. 52). This implies that noise levels rather play only a modest role in the annoyance response (e.g. Van Kemp and Van den

Berg, 2021; Schultz, 1978, p. 378).

To better understand the particularly annoying nature of wind turbine noise, it has become increasingly recognized in the health-based epistemic community that the special sound characteristics of wind turbine noise (e.g. amplitude modulation), and notably "non-acoustic factors", matter. Indeed, it has been estimated that "non-acoustic factors may explain up to 33%

of the variance" in noise annoyance studies (Guski, 1999). Hereby, "reducing the impact of wind turbine sound will profit from considering other aspects associated with annoyance" (Van Kamp and Van den Berg, 2021, p. 25), i.e. a range of contextual and personal factors in addition to actual sound exposure levels. These are things such as visual aspects and demographics, personal, social, political and economic aspects (Van Kamp and Van den Berg, 2021, p. 16). Some of these non-acoustical factors were identified in the social acceptance literature (Wolsink et al., 1993) (see Social Acceptance

section). To account for these variations of the parameter – "polluting" the dose-response calculations - other metrics for annoyance have also been presented, for example, "aggregated annoyance" in the CNHS study (Michaud et al., 2016a), which attempts to take non-acoustic aspects into account.

The often personal and contextual non-acoustic factors that are less quantifiable and calculable have continued to produce

uncertainty in the interpretation of the dose-response graphs. They have limited the graphs' ability to provide noise limit guidance across very different local community settings, as different communities' annoyance levels vary widely despite being exposed to the same noise levels (e.g. Michaud, 2016b). This uncertainty also leads to debates in the health community about the logic of the "10% highly annoyed rule" to set a unitary noise level limit (dB(A) number), as the percentage of highly annoyed changes across studies and over time, i.e. with the evidence base (INCE Conference, 2021).






Another key example of inconclusive attempts at taming the unruly nature of the epistemic object of wind turbine noise are the publications that came out of the Danish health study on wind turbine noise (e.g. Poulsen et al., 2018). Funded by the Danish government with high hopes that the study could provide "objective proof" on (the lack of) health impacts (cardiovascular disease and other diseases related to stress and sleep deprivation), the study was in the end unable to
establish with certainty either that there were or there were not health effects from wind turbine noise, and the researchers behind the study recommended further research (Danish Agency of Public Health (Sundhedsstyrelsen), 2019; Poulsen et al., 2019). Thus, in a sense, the cancer study has produced evidence, or, rather "arguments" both for proponents of and opponents to wind power, sparking even more controversy.

### 4.3 Noise in the Social Acceptance community

Social science scholars in the field of social acceptance of renewable energy technologies (e.g. Batel, 2020; Ellis and Ferraro, 2016; Wüstenhagen et al., 2007) have also dealt with wind turbine noise. This epistemic community centres on acceptance as the social mediator for implementing renewable technologies and aims at highlighting the biggest obstacles and opportunities for achieving this acceptance. We unfold below how noise has been treated as an obstacle for acceptance in this field.

### 4.3.1 The dose-response effect in social acceptance

Within social acceptance, noise is never studied in isolation. Instead, several different factors that might influence acceptance are studied together often using surveys. When noise is considered in these surveys, the previously described dose-response studies and the linked concept of noise annoyance often act as a considerable source of inspiration. Several articles in the field refer directly back to the pivotal dose-response study conducted by Pedersen and Waye (2004), as a way of signalling
the special character of wind turbine sound and its potential impact on acceptance (e.g. Cashmore et al., 2019, p. 1113; Haggett, 2012; Hill and Knott, 2010, p. 157; Walker et al., 2015, p. 359). When conducting surveys on residents' perceptions of turbines, the social acceptance literature often also mimics the questionnaires of dose-response studies by asking about noise annoyance (Baxter et al., 2013; Brudermann et al., 2019; Frantál et al., 2017; Wolsink et al., 1993). While the concept of noise annoyance dominates, other ways of asking about noise in surveys do occur (see e.g. Dällenbach and Wüstenhagen
2022, p. 4f; Kontogianni et al., 2014, p. 174). It should also be noted that there have been a few attempts at grasping the relations between noise and acceptance qualitatively where reactions to noise are understood as socially experienced and culturally contingent (Eun-Sung Kim and Chung, 2019; Eun-Sung Kim et al., 2018; Haggett, 2012; Batel and Devine-Wright, 2021). Yet, such qualitative approaches have not gained a lot of traction in the social acceptance-based epistemic community with regard to the issue of noise.





### 4.3.2 The relationship between noise and visual aspects

Since the earliest surveys on local acceptance of wind turbines, two factors have stood out as especially influential on acceptance: visual/landscape impact and noise (see e.g. Bosley and Bosley, 1988; Pasqualetti and Butler, 1987). Yet, over time a consensus that visual or landscape factors are more influential on acceptance than noise has developed (Wolsink, 2007a, 2007b). This argument especially dates back to a study conducted by Wolsink and colleagues in the early 1990s (Wolsink et al., 1993) that only found a weak relationship between sound pressure levels and noise annoyance. On the other hand, the researchers found that the degree of visual intrusion was affecting the level of noise annoyance considerably (Wolsink et al., 1993; Wolsink and Sprengers, 1993). That visual factors are dominating aural factors in determining local acceptance was further backed by Wolsink (2000) in a highly cited study. Through a combination of factor analysis and linear regression this study showed that resistance toward specific wind turbines is most affected by the general attitude towards wind energy followed by the visual assessment of turbines. It was also found that noise had a significant yet smaller impact on resistance. Further, it was found that the visual factor had an indirect effect on resistance by greatly influencing the general attitude towards wind energy, while noise had no significant impact on this (ibid., p. 54f).

### 4.3.3 Other factors mediating between noise and acceptance

Apart from the wind turbines' fit with the landscape, matters of justice are also regularly highlighted as important for local acceptance – both in terms of how costs and benefits are shared (distributional justice) and in terms of the fairness of the planning and decision making process (procedural justice) (Wüstenhagen et al., 2007, p. 2685). Among studies focusing on fairness of planning, relations to noise annoyance have also been identified: For instance, several authors are pointing out that noise annoyance is less prevalent among those who benefit economically from wind turbines (Janssen et al., 2011; Tabassum et al., 2014, p. 276). Further, a comparative study across the U.S. and Europe found that perceptions of the fairness of the planning process and whether the planning process was experienced as stressful or not had some effect on the noise annoyance once the wind farms were in operation (Hübner et al., 2019; Pohl et al., 2018). Dällenbach and Wüstenhagen (2022) hypothesized that local noise concerns would peak just beyond the borders of the municipality where a wind farm under study was planned due to lack of inclusion of residents of neighbouring municipalities in the planning processes. Though their results were mixed, the authors found some indices of this tendency, and they suggested that thinking about procedural and distributional justice is key for the acceptance of wind farms (ibid., p. 9ff).

### 4.3.4 Noise as a proxy for other concerns

Given that noise annoyance seems to be highly influenced by both visual and justice aspects, noise annoyance's effect on acceptance may be nothing more than a spurious relationship. Yet, this makes it all the more puzzling that noise is often among the most frequently debated issues in controversies over local wind farms. A common explanation of this in the literature is that in most planning systems it is easier to complain about noise than landscape aspects or procedural justice.





Hence, noise complaints function as a proxy for other concerns (Hill and Knott, 2010, p. 167; van der Horst, 2007, p. 2711; Wolsink, 1989, p. 12; 2000, p. 56).

### 4.3.5 Noise pollution polluted by other nuisances - noise as a social issue

It is a point in itself that this section started with noise annoyance but ends with other nuisances such as visual/landscape
annoyance and stress from the planning process: Within the social acceptance epistemic community, the dose-response relationship between wind turbine noise and noise annoyance has undergone several re-examinations adding social complexity to the context in which people experience noise annoyance.

Adding this complexity changes the flavour of noise annoyance: where it is framed as a technical or health issue in other
epistemic communities, noise here becomes a social issue relating to the organisation of the planning process and the landscape impact affecting aesthetics and place identities. Overall, the social acceptance literature moves away from a direct relationship between noise as sound pressure and annoyance, which complicates the established relations between dose and response in the epistemic community of health.



## 5 Discussion


Here, we distil our findings and discuss what we consider are the implications, in the light of this study. It is clear from our analysis that the three epistemic communities understand and treat noise as different things. Table 1 summarizes the different understandings, by distilling the following aspects: 1) historical background of the study of noise, 2) the understanding of noise, 3) methods of detection and measurement, and 4) solutions presented to try to address the issue of noise.


**Table 1: Summarising the many understandings of noise.**

|  | Technical | Health | Social Acceptance |
|---|---|---|---|
| **Historical background** | Study of turbulent flow based on aircraft wings. | Grew out of concern for environmental noise | Stems from an interest in understanding people's degree of acceptance of wind farms |
| **How is noise understood?** | A physical phenomenon – vibration of air particles caused by turbulence | First health concern was direct damage on ear drum but later understood as a possible cause of several health effects such as annoyance and thus stress | As one of several factors including visual impact, issues of justice and fairness of planning, that influences the acceptance of wind farms |
| **How is noise found?** | Measured by microphones and assigned a volume level in dB(A). generation and propagation modelled by digital tools. | Through surveys or laboratory studies detecting the degree of annoyance caused by  various noises. | Through surveys and interviews |
| **How is noise being treated or solved?** | Treated as 'unwanted'. Main effort is in reducing the volume i.e. lowering the dB(A) | By recommending noise limit values to regulators | By assessing the extent to which noise affects the acceptance of wind farms |

The table shows that, in short, noise is not only not understood as the same thing, it is not the same thing across the three

communities; instead, the different understandings construe and enact noise as different things, revealing it as a variable and contested 'thing', that appears to defy a straightforward 'solution'. One way we have tried to show this is in Fig. 3, that reveals how noise is a slippery and unruly construct that undergoes a curious shift – mutation (Knorr-Cetina, 1999) - across the different epistemic communities: even though it is 'noise' that is ostensibly the object of enquiry in all three, the





knowledge object of noise exists, we show, in a variety of (simultaneous) forms, also enabling the binding together of
collectives such as epistemic communities (Knorr-Cetina, 1999, p. 16) in different ways.

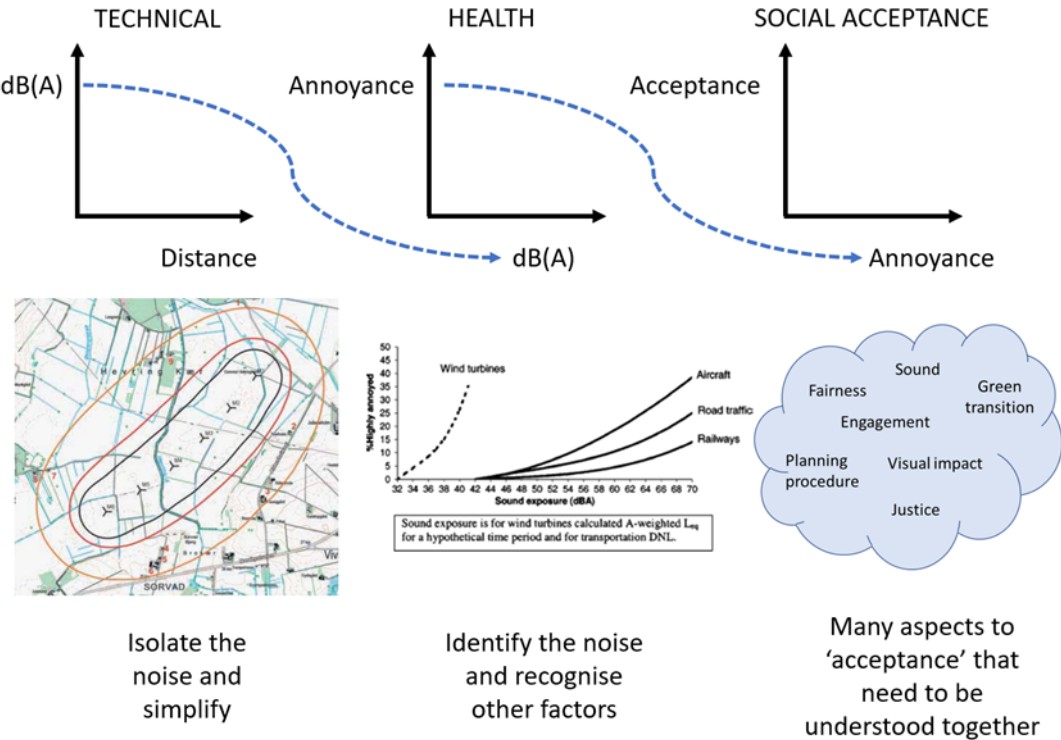

**Figure 3: Shifting dependent-independent variables in the understanding of noise (map from Hevring Ådal wind farm EIA, 2012)**

First, in the technical epistemic community, noise is sound pressure levels measured using the dB(A) scale. In our manner of
analysis, we can consider that this constitutes the dB(A) level as the dependent variable, and the problem to be 'fixed'. By
reducing the problem to a measurable metric such as the dB(A), the primary solution (in our analysis, the independent
variable) has been the introduction of various technical fixes that can reduce the noise level (e.g. dinotails) such that the
distance to the required noise level contour is less than to citizens' properties.

Second, and moving on to the health-based epistemic community, noise is (measured through and translated into) annoyance
levels (% highly annoyed people). That is, construed as a dependent variable, annoyance is a product of the dB(A) level
(sorted and purified from "non-acoustic factors"), which means that the solution becomes associated with reduction of dB(A)
levels through influencing noise regulations to take people's health (annoyance) into account.





Finally, in the social acceptance epistemic community, noise becomes one of several independent variables for determining the level of social acceptance that is the dependent variable of this community. With inspiration from the health epistemic community, noise is often operationalized as noise annoyance that can be asked about in surveys. Researchers in this field rarely have the tools to measure or calculate noise doses in dB(A) and hence dB(A) levels tend to fade from their analysis.

Through this analysis above, it becomes clear that, while the three communities are all studying "noise", the challenge is that noise is not the same thing to each of them. It is interesting to reflect on how noise seems to shift in quality as it travels between the technical, health and social acceptance communities. Thus, if noise is not the same thing, then the issue surrounding noise will be different, and the approach and solutions will also be different. What is problematic is that because of the curious shift that noise undergoes, the technical community think that by reducing the noise level in dB(A), then they
are addressing the issue of social acceptance. What they really are addressing is a highly isolated and technical issue of how to reduce the volume of noise, but its relation to social acceptance is a complex and entangled path. Similarly, the health community consider that they are addressing the issue of social acceptance via the proxy of annoyance. However, the understandable inability to meaningfully encompass non-acoustical factors and the adherence to the dB(A) metric, mean that they tend to be addressing the needs of regulation, rather than being able to respond to peoples' concerns about living in the
proximity of wind farms.

Why might this be the case, and what could be done to move forward?

**5.1 Dealing with scientific uncertainty**

In this final part of the paper, we would like to present some considerations that result from our approach to this study and to
shed some light on why we are where we are with the issue of wind turbine noise. We consider that the different ways of handling and treating the somewhat unruly nature of noise across the three communities can be related to the role of uncertainty in science. It is clear that there is continuous and considerable work to distil and compartmentalise the noise issue into either the metric of dB(A) or the metric of annoyance levels. From our perspective, this reflects attempts at stabilising the uncertain into quantifiable "risks", that is, construing unruly noise as a governable agreed-upon (framed)
object of inquiry. Of the three epistemic communities, the technical community seems to be the most uncomfortable with uncertainty, with repeated attempts to solve the noise issue quantitatively and relating solely to acoustic factors. The health-based community, in turn, acknowledges, to some extent, uncertainty, e.g. by using survey methods and a statistical approach ("percentage of highly annoyed"). Furthermore, there is a growing appreciation of the role of non-acoustic factors, although they are difficult to handle and complicate the rather clear dose-response models. The social acceptance
community, however, appears the most comfortable with uncertainty, recognising that the context of the issue is of primary importance and that it is impossible and indeed, undesirable, to disentangle it from other factors; that is, here acoustic and non-acoustic factors are entwined.



We can now return to the somewhat paradoxical situation we outlined in the introduction, whereby significant investments
are still being made into quantifying and reducing the sound "dose" (e.g. the Poul la Cour wind tunnel) when the epistemic
communities around social acceptance and health have been saying for years that it is not the absolute sound level that
determines people's responses to noise (Taylor and Klenk, 2018; Müller et al., 2023; Thorne, 2011). How did we get here?

The continued impetus on noise levels and the dB(A) metric in the technical and health-based communities may be best
explained by their epistemic history (see Nyborg et al., forthcoming), or what has been referred to as the relational effect of
institutional and spatial arrangements that "stick" and make it hard to do things otherwise (Eyal, 2013). That is, both the
technical and health-based epistemic communities have historically been closely related to informing policies, regulations,
and standards on noise limits such as the ISO standard for noise, IEC standards, and national regulations. They have, over
time, gained agency to impact policies and regulations through the power of their numbers, quantification and handling
uncertainty. Noise regulation, by nature, relies on "scientific facts" and tries to set a more or less measurable limit below
which it is most likely that people are protected, as regulations and standards are there to set the frame for industrial
development while not harming public health.

With this historical background in mind, we recommend that future studies should look into issues of agencies of different
epistemic communities to impact policy and regulation, and the role of their devices, such as the powerful role of the dose-
response model has had to impact policymaking in the case of noise. Through this, the lack of the voice of the social
acceptance community in informing policymakers can be explored, with the realisation that their broader attention to matters
other than noise level limits (e.g. visual impact, fairness and justice) cannot be, and also have value in not being, distilled
down into the metric of the dB(A) and that uncertainty is a valuable attribute of the contextual nature of wind power
development.

We thus return to the central premise of the Co-Green project, also outlined in the introduction, as a means to suggest how to
move forward. In Co-Green, we have coined the perspective of "technification", that is, reducing and taming the issue of
noise into measurable and technical solutions. Our research indicates that this very exclusion of less measurable attributes in
the technification process can, at times, backlash, stirring up controversy. Just as, for example, the Danish Heath Study on
wind turbine noise mentioned earlier did not "tame" the issue, but enhanced (and disappointed) expectations of enhanced
public engagement and expectations of bringing scientific certainty. Indeed, scholars in the STS field (e.g. Latour, 1987,
2005; Callon, 1980; Callon et al., 2009) have often been highlighting that trying to solve controversies by providing more
techno-scientific knowledge on an issue is often counter-productive, and can even spark more controversy. Our call here,
therefore, is for a more nuanced engagement between the scientific communities, which could present a means to better
account for the more "uncertain" qualitative issues of noise, potentially providing a basis for more balanced scientific policy





and regulation input. We are fully aware, though, that this is no easy task: our study here indicates how the different understandings of noise can make such interdisciplinary dialogue difficult (see Nyborg et al. forthcoming). We still, though, encourage enhanced reflexivity over how noise and the uncertainty associated with noise, is not just there a priori but is
construed by epistemic communities and made through their respective tools and devices. With this awareness of each other's epistemic history, our hope is that this paper can positively contribute to a more informed interdisciplinary dialogue.

## 6 Conclusion

In our study we set out to explore the question of how noise is understood by different scientific literatures and the effects these different understandings have on the solutions to the issue of noise that they propose. Based on our narrative review,
this paper gives an account of how wind turbine noise over time has been construed as different epistemic objects by three different epistemic communities ("Technical", "Health" and "Social Acceptance"). Our analysis has shown that, as a consequence, "the problem" of wind turbine noise has been construed differently, leading to solutions that maybe address something different than how "acceptable" people consider wind turbine noise.

Whilst this finding is not necessarily revolutionary from a social science perspective (e.g. Mol, 2024; Knorr-Cetina, 1997, 1999; Latour, 2005), the issue of noise from wind turbines is a highly relevant and rich case. It is therefore worth summarising the communities' understandings here, to crystallise their differences: For the technical community, noise is something that can be measured and modelled, with the expectation that its impact on people can be reduced to a quantifiable estimate of dB(A). Hereby, there is the assumption that by quantifying and reducing the noise level, the risk of
opposition to wind energy as well as revenue loss, is reduced. For the health community, a statistical approach translates the noise dB(A) level into the risk of a certain percentage of highly annoyed people, enabling the setting of a limit that can be relied upon for informing policy and regulation, providing a quantifiable threshold below which the risk of adverse health effects is reduced to an acceptable level. And, finally, for social acceptance, noise is just one feature of the complexity of the issue of wind power development. Noise, visual impacts, procedural fairness, etc., are all entangled and the complexity is
something that is not quantifiable but considered something that can usefully inform research.

We have also teased out the issue of the approach to noise of these communities, seen through the perspective of handling uncertainty, which has led us to explore the issue of the continued funding into quantifying and reducing the noise level, when there are significant voices that question whether it is noise level that determines people's responses to the issue of
wind turbine noise. In many ways, our research shows that this continued focus on the technification of noise may well be the reason why it continues to be a difficult issue to resolve and why it still causes controversy despite the considerable investments made in trying to tame it.



We thus conclude with a call for two further avenues of research and study to move forward on the issue of controversy over wind turbine noise. The first is to use this newly uncovered reflexivity about the multiple understandings, and uncertainties, of noise to provide a stronger basis for interdisciplinary engagement and research between the three epistemic communities, to clarify how the issue of noise is being problematised, and thus to align efforts as to how to solve it. The second, is a call for exploring the roles of the models and devices that are used by the communities to inform policy and regulation, in an effort to see how scientific communities that do not work with quantifiable aspects of wind energy can usefully play a role in framing the issue of promoting complexity and an understanding of uncertainty in science-based regulation.



**APPENDIX A: Search strings used for literature review.**

Databases searched: WoS, Scopus and PubMed.


**Technical perspective**

(("wind turbine*")  OR ("windmill") OR ("wind farm") OR ("wind energ*") OR ("wind power")) AND (noise OR sound))
AND (source OR generation OR mitigation OR propagation OR modelling OR measurement OR tonality OR "amplitude
modulation" OR "low frequency" OR "infra sound")


**Regulatory literature**

(("wind turbine*")  OR ("windmill") OR ("wind farm") OR ("wind park") OR ("wind energ*") OR ("wind power")) AND
((noise) OR (sound))) AND ((regulat*) OR (standard* OR IEC) OR ("polic*"))

**Health perspective**

(("wind turbine*" OR "windmill*" OR "wind farm*" OR "wind park*" OR "wind energy*" OR "wind power*") AND (nois*
OR sound*) AND (health* OR annoy* OR percep* OR sleep*))

**'Social Acceptance' perspective**

(("wind turbine*"  OR  "windmill*"  OR  "wind farm*"  OR  "wind park*"  OR  "wind energy*"  OR  "wind power*"  AND
nois*  OR  sound*  OR  annoy*  AND  opinion*  AND NOT  opposition  OR  complain*  OR  resistance  OR  nimby*  OR
acceptance))





**APPENDIX B: List of anonymized interviews.**

|      | **Type of expert** | **Date of interview** |
|------|--------------------|-----------------------|
| B1   | Senior researcher in wind turbine noise | 10.03.2021 |
| B2   | Senior researcher in wind turbine noise | 15.04.2021 |
| B3   | Acoustics engineer in measuring noise | 06.05.2021 |
| B4   | Professor in wind turbine noise | 12.05.2021 |
| B5   | Experimental Psychologist and Behavioural Scientist | 11.06.2021 |
| B6   | Consulting Engineer | 11.06.2021 |
| B7   | Noise specialist for public body | 14.12.2021 |
| B8   | Industry Association | 01.02.2022 |
| B9   | Environmental Geography | 25.02.2022 |
| B10  | Industry Association | 25.02.2022 |
| B11  | Acoustics engineer in measuring noise | 12.11.2022 |
| B12  | Acoustics researcher | 23.11.2023 |



**APPENDIX C: Example timeline used in interviews.**

This example was used in the interviews with experts from the health community.






**Author contribution:**

Kirkegaard J., set the framing of the paper, wrote the introduction, discussion and conclusions, and edited the whole paper.
Cronin, T., made the review of the technical literature and contributed to the editing of the whole paper. Nyborg, S., made the review of the health literature and reviewed the final paper. Frantzen, D., made the review of the social acceptance literature and reviewed the final paper. All authors contributed to conducting the expert interviews and the paper's analysis.

**Competing interests:**


At least one of the (co)authors is a member of the editorial board of Wind Energy Science.

**Acknowledgements:**

The authors would like to thank Professor Maja Horst for early discussions concerning the conceptual framing of the Co-Green project and this paper. We would also like to thank Emil Nissen who helped with interview transcriptions, making the timeline props for interviews, and comments generally throughout the course of this paper.

We would like to acknowledge and thank the funding of the Co-Green project ("Controversies in the green transition: The
case of wind turbine sound and its politicisation") by the Independent Research Fund Denmark, Grant No. 0217-00229B.



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
