# Peer review of "The multiple understandings of wind turbine noise: Reviewing scientific attempts at handling uncertainty."

_Wind Energy Science, 2024_

## Editor Comment (EC2)

Comments on: The multiple understandings of wind turbine noise: Reviewing scientific attempts at handling uncertainty.

My main comment is that the discussion of the fundamentals of sound and the characteristics of noise from wind turbines is insufficient for what the paper is trying to do.  The purpose of the paper seems to be to emphasize the importance of the non-technical aspects of noise while deemphasizing, close to denigrating, the "hard science"/technical aspects of noise.  By this point in time it is clear from many studies that the sound pressure level at locations in the vicinity of wind turbines, measured in dB(A), is insufficient to explain resistance to wind turbines.  The paper does a good job at summarizing some of the key observations in this regard.  What is lacking is describing the nature of wind turbine sound and relating it to that resistance.  The paper should begin with a review of what sound (and thus noise) are, how they can be quantified, how wind turbines produce sound and how the sound is propagated.

Fundamentally, sound (in air) is the result of pressure variations in the air, and this can originate in a variety of ways.  These vibrations are characterized by their frequency and the amplitude of the various frequencies.  The vibrations can span a wide range of frequencies, from below audible (20 Hz, i.e infrasound) to above audible (20 kHz, i.e. infrasound), and the sound pressure amplitudes can also span a wide range, from just barely audible (20 µPa, i.e. $2 \times 10^{-5}$ Pa) to above the pain threshold (200 Pa).  Because of the wide range of amplitudes and for historical reasons values are reported on a log scale, referenced to 20 µPa, which provides a more compact range, for example 0 dB to 140 dB instead of 20 µPa to 200 Pa.  It is also the case that human auditory response depends on frequency, so weighting scales, denoted A, B, C, G and Z, have been developed.  The most commonly used scale is the A weighting scale; its effect is to attenuate the response for frequencies below 1000 Hz, so the reporting is given in units of dB(A).  A summary measure of the total effect of all the vibrations, SPL_av, is given by integrating the amplitudes at all frequencies over all frequencies.  When the weighting is included, that apparent total will be reduced below the value that would be obtained without the weighting.  It is important to note that a given average sound power level can correspond to different amplitudes at different frequencies, so when frequencies are relevant to the response, the SPL_av is only partially relevant and may even be misleading.

In common usage today, the SPL_av, is measured by an instrument which does the integrating and filtering over all relevant frequencies and reports out one value over some prescribed sampling time.  This method of reporting is done for historical reasons since it has heretofore not been convenient, or perhaps necessary, to collect more detailed information.  In the case of wind turbines, amplitude as a function of frequency, and also amplitude as a function of time ("amplitude modulation"), may well be interesting and in fact such information may now be readily obtained.  The authors should consider this possibility.

The types of wind turbine noise and the ways in which turbines produce sound are also relevant but were not discussed in the paper.  There are at least four types of noise: tonal, broadband, low-frequency, and impulsive.  The characteristics of the noise at a given location are a function of the turbine itself as well as the environment in which it operates.  For example, upwind vs. downwind rotors have different noise characteristics.  Higher blade tip speeds result in significantly higher noise production than slower tip speeds.  Rotors can be designed to operate with lower tip speeds but then the total blade area (and thus blade mass and so blade costs) will be higher, so there are trade-offs to be considered.  As has been widely reported, wind turbines may produce a thumping or whooshing sound; these have particular

frequency and temporal features which are not apparent from the SPL_av data.  Wind shear and atmospheric stability may also affect the noise produced and how it is perceived.  The occurrence and characteristics of the sound may be weather dependent, and variations may occur in an apparently random and hence non-predictable manner.

The frequency characteristics of wind turbines, including low-frequency sound and infrasound, has been discussed by a variety of authors, but were not included in the paper- they should be.  Examples include: Leventhall, G. (2009). "Low Frequency Noise. What we know, what we do not know and what we would like to know." Journal of Low Frequency Noise, Vibration and Active Control, 28(2). 79-104 and Leventhall, G. (2006, June). "Infrasound from wind turbines - fact, fiction or deception." Canadian Acoustics, v 34, n 2, p 29-36.  Another report that has potentially useful references is the Wind Turbine Health Impact Study (https://www.mass.gov/doc/wind-turbine-health-impact-study-report-of-independent-expert-panel/download).

The use of predicted average sound pressure level may be less than ideal from a number of perspectives.  For regulators to use SPL_av as a basis of permitting may compel turbine designers and project developers to focus on reducing that average value.  If the frequency characteristics are actually more important, then the standards and regulations should be updated accordingly.  Turbine designers would have an additional impetus to fine-tune designs, perhaps to even consider "noise cancelling".  In summary, there may well be considerations other than purely technical regarding the wind turbine noise conundrum, but it should not be assumed that the average sound power level is the only relevant technical issue.

---

## Editor Comment (EC3)

[revised manuscript text omitted]
 re̶d̶u̶c̶i̶n̶g̶ ̶t̶h̶e̶ ̶n̶o̶i̶s̶e̶ ̶o̶f̶ ̶t̶u̶r̶b̶i̶n̶e̶s̶ (true but so what?) ̶(̶e̶.̶g̶.̶ Poul la Cour wind tunnel) when the epistemic
communities around social acceptance and ̶s̶o̶u̶n̶d̶ ̶p̶e̶r̶c̶e̶p̶t̶i̶o̶n̶ ̶a̶r̶e̶ ̶s̶h̶o̶w̶i̶n̶g̶ rs that it is not the absolute sound level that
determines people's responses to noise (Taylor and Klenk, 2018; Müller et al., 2023; Thorne, 2011). How did we get here?

The continued imp̶o̶r̶t̶a̶n̶c̶e̶ ̶a̶n̶d̶ ̶i̶m̶p̶a̶c̶t̶ ̶o̶f̶ ̶t̶h̶e̶ ̶t̶e̶c̶h̶n̶i̶c̶a̶l̶ ̶a̶n̶d̶ ̶h̶e̶a̶l̶t̶h̶-̶b̶a̶s̶e̶d̶ ̶e̶p̶i̶s̶t̶e̶m̶i̶c̶ ̶c̶o̶m̶m̶u̶n̶i̶t̶i̶e̶s̶ may be best

[The metric of dB(A) was set in an earlier time when measurement of sound according to frequency was more difficult. Other metrics could now be readily developed which would be relevant for all concerned.]

explained by their e̶m̶b̶e̶d̶d̶e̶d̶n̶e̶s̶s̶ ̶i̶n̶ ̶a̶n̶d̶ ̶t̶h̶e̶i̶r̶ ̶i̶n̶t̶e̶r̶c̶o̶n̶n̶e̶c̶t̶i̶o̶n̶s̶ nal effect of
institutional and sp̶a̶t̶i̶a̶l̶ ̶a̶r̶r̶a̶n̶g̶e̶m̶e̶n̶t̶s̶ ̶t̶h̶a̶t̶ ̶"̶s̶t̶i̶c̶k̶"̶ ̶a̶n̶d̶ ̶m̶a̶k̶e̶ ̶i̶t̶ ̶h̶a̶r̶d̶ ̶t̶o̶ ̶d̶o̶ ̶t̶h̶i̶n̶g̶s̶ ̶o̶t̶h̶e̶r̶w̶i̶s̶e̶ ̶(̶L̶y̶a̶l̶,̶ ̶2̶0̶1̶5̶)̶.̶ ̶T̶h̶a̶t̶ 
[revised manuscript text omitted]

---

## Author Response (AR1)

Reviewers' comments and authors' responses:
The multiple understandings of wind turbine noise

| | Reviewer's Comment | Authors' Response |
|---|---|---|
| **RC1** | | |
| | Overall feedback: The manuscript deals with a very interesting and timely topic of wind turbine noise. Providing a critical, social science view on what is sound and what is noise and how science is involved in production of the phenomenon of noise and how it can be tackled, The MS is written in a way that is accessible to broad readership. In particular, the authors did a good job at positioning themselves in the 'shoes' of technical audience and explained very well their approach and how this manuscript could be useful for them. | Many thanks for the positive reception of our MS, and for the helpful comments and suggestions. |
| | Your paper as such can help a lot the technical audience to observe the ways in which they tackle the problem of wind turbine sound/noise- in a way you hold a mirror for the- but I believe it is also useful to the policy makers and people involved in legislation as well as management of wind farms. Perhaps you could also mention this audience in your work and conceive recommendations for how this MS is relevant for their work. | We have added these audiences and a brief suggestion as to how this paper could be relevant for their work. |
| | Abstract:

Various communities of scientists, here understood as epistemic groups. Maybe mention that these are groups working in different domains and what are these? Divided into single disciplines or how? | Regarding the abstract, we have restructured it so that our definition of the communities of scientists/epistemic is spelled out from the outset; hereby we will tie the technical literature to the engineering discipline, health-based to medicine/psychology, and social acceptance literature to the social science discipline. |
| | Introduction:

You start by arguing that sound becomes 'politicized noise'. Can you explain what do you mean by this what seems to be a process of politicization | In the revision, we have decided to omit the notion of "politicized" noise as we do not use this term more than once in the MS. Instead, we have reformulated it as "controversial" (that is, contested) noise. |
| | In the line 35 you discuss the controversies and issues around wind turbine sound, perhaps add reference to prior research that discussed it? | Regarding your comment (line 35) on lacking references regarding controversies and issues around wind turbine sound, we have added some relevant references. |
| | You state: "These issues and their implications for wind farm development are typical of those seen in Denmark, and there has been significant funding of ambitious projects to resolve the issue." Maybe don't say ambitious projects but make it more specific? Research projects? | Thanks for your comment on the lack of specificity on the type of projects funded to inquire into wind turbine noise. We have omitted the notion of 'ambitious', instead qualifying them according to their type (research, commercial). |
| | You state " how concepts such as noise are co-produced by the scientific communities that form around them" this is the first time you mention the idea of co-production, maybe good to explain what you mean when you use the concept? Or use a simple term instead.

Would it not be useful to discuss the public perceptions of noise? How they are different from the scientific ( and as you argue multiple) understandings of sound/noise. | Regarding the notion of co-produced noise, we have decided make a reformulation in order to simplify. We now state that different communities create/produce their own understanding of what noise is, and that this has implications for how they deal with it. |

| | | |
|---|---|---|
| | | In other publications, we have directly addressed public perceptions of noise, but the scope of this paper is to focus on how the scientific literatures have looked at the issue of noise; here, the social acceptance literature is the one engaging most directly with public perceptions. We have added a sentence in the paper on this scoping issue (introduction). |
| | **Results**
Your results are very interesting but mostly describe the state of art in relation to literature and not views expressed by the experts tehmselves. On this point, I wonder what was the role of your interviews in the data analysis as you do not seem to draw directly on this data in your results section. Would quotes from your interviews help to enrich the results and provide expert views? | Thanks for this comment. We have now added an explanation of our use of interview data in the methods section. Our main data for this paper has been the literature reviewed (text analysis). We conducted interviews with the purpose of corroborating, cross-checking and verifying our analysis of these texts. Meanwhile, we consider using quotes from these expert interviews out of scope for this paper; we elaborate more on these in other papers. |
| | As a last point of thought, I am wondering how your analysis what noise is would be if you also took into consideration animals? | Last, we want to thank you for commenting on the issue of animals and their perception of noise. We believe the three reviewed literatures do not deal much with this and so we have not treated it in the MS. We hope that future research could look into this overlooked aspect of wind turbine noise. |
| **EC2** | Comments on: The multiple understandings of wind turbine noise: Reviewing scientific attempts at handling uncertainty.

My main comment is that the discussion of the fundamentals of sound and the characteristics of noise from wind turbines is insufficient for what the paper is trying to do. The purpose of the paper seems to be to emphasize the importance of the non-technical aspects of noise while deemphasizing, close to

denigrating, the "hard science"/technical aspects of noise. By this point in time it is clear from many studies that the sound pressure level at locations in the vicinity of wind turbines, measured in dB(A), is insufficient to explain resistance to wind turbines. The paper does a good job at summarizing some of the key observations in this regard. What is lacking is describing the nature of wind turbine sound and relating it to that resistance. The paper should begin with a review of what sound (and thus noise) are, how they can be quantified, how wind turbines produce sound and how the sound is propagated.

[…]
The use of predicted average sound pressure level may be less than ideal from a number of perspectives.

For regulators to use SPL_av as a basis of permitting may compel | Thank you for taking the time to read our paper and for the detailed suggestions to add to our MS.
The overall comment concerns the lack of fundamentals of sound and noise characteristics of wind turbines, and the reviewer has kindly provided a number of technical details. We have added some extra details about these aspects where they enhance a particular point (see below). However, given the scope and aim of this paper, namely to offer an account of different understandings of noise, and to do this based on social science methods and perspectives, we cannot go into too many details with the technical aspects. We consider that the fundamentals of sound and noise are well documented in scientific text books and the technical literature, and our aim is not to reiterate these. We have added a further explanation of this in the revised text.
Indeed, we argue that through understanding how the different scientific |

| | | |
|---|---|---|
| | turbine designers and project

developers to focus on reducing that average value. If the frequency characteristics are actually more

important, then the standards and regulations should be updated accordingly. Turbine designers would have an additional impetus to fine-tune designs, perhaps to even consider "noise cancelling". In

summary, there may well be considerations other than purely technical regarding the wind turbine noise conundrum, but it should not be assumed that the average sound power level is the only relevant technical issue. | communities see 'noise' as something different, we highlight why resolving the issue of noise continues to be difficult. We then go on to suggest that these communities could benefit from engaging more with each other, resulting in fresh input to, for example, technical research, and aligning efforts to solve the noise issue.

In the revised text we have addressed the following additional technical aspects, to the extent that we consider they augment the paper: frequency and pitch of sound, possibilities for other metrics than dB(A), the average sound level is not the only technical aspect under research, dependency on wind shear, landscape effects, weather impacts, and possibilities for frequency and time-dependent regulations. |
| EC3 | Comments made in text | Thanks for the comments provided to our MS, these are all very helpful. We have corrected the text accordingly, regarding language clarifications/definitions. |
| | Overall comment | Concerning the overall comment on the 'major omission' – namely the lack of "fundamentals of sound" (p. 16) – we will certainly add some extra lines about these aspects where they enhance a particular point. (Please see the additions made in the comment above). However, given the scope and aim of this paper, namely to offer an account of different understandings of noise, and to do this based on social science methods and perspectives, we cannot go into too many details with the technical aspects. We consider that the fundamentals of sound and noise are well documented in scientific text books and the technical literature, and our aim is not to reiterate these. We have added a further explanation of this in the revised text.
Indeed, we argue that through understanding how the different scientific communities see 'noise' as something different, we highlight why resolving the issue of noise continues to be difficult. We then go on to suggest that these communities could benefit from engaging more with each other, resulting in fresh input to, for example, technical research, and aligning efforts to solve the noise issue. |

---

## Editor Decision (ED1)

**ANONYMOUS REVIEWER COMMENTS NOT INCLUDED IN PUBLISHED VERSION**
The **anonymous reviewer** was not happy that the authors apparently ignored much of what I said, but I guess it is OK to publish the paper if additional comments are included alongside, as you indicated may be possible.  In anticipation of that, I have slightly edited my earlier comments and will paste them here.

My main comment is that the discussion of the fundamentals of sound and the characteristics of noise from wind turbines is insufficient for what the paper is trying to do.  The purpose of the paper seems to be to emphasize the importance of the non-technical aspects of noise while deemphasizing, close to denigrating, the "hard science"/technical aspects of noise.  By this point in time it is clear from many studies that the sound pressure level at locations in the vicinity of wind turbines, measured in dB(A), is insufficient to explain resistance to wind turbines.  The paper does a good job at summarizing some of the key observations in this regard.  What is lacking is describing the nature of wind turbine sound and relating it to that resistance.  The paper should begin with a review of what sound (and thus noise) are, how they can be quantified, how wind turbines produce sound and how the sound is propagated.

Fundamentally, sound (in air) is the result of pressure variations in the air, and this can originate in a variety of ways.  These vibrations are characterized by their frequency and the amplitude of the various frequencies.  The vibrations can span a wide range of frequencies, from below audible (20 Hz, i.e. infrasound) to above audible (20 kHz, i.e. infrasound), and the sound pressure amplitudes can also span a wide range, from just barely audible (20 µPa, i.e. $2 \times 10^{-5}$ Pa) to above the pain threshold (200 Pa).  Because of the wide range of amplitudes and for historical reasons values are reported on a log scale, referenced to 20 µPa, which provides a more compact range, for example 0 dB to 140 dB instead of 20 µPa to 200 Pa.  It is also the case that human auditory response depends on frequency, so weighting scales, denoted A, B, C, G and Z, have been developed.  The most commonly used scale is the A weighting scale; its effect is to attenuate the response for frequencies below 1000 Hz, so the reporting is given in units of dB(A).  A summary measure of the total effect of all the vibrations, $SPL_{av}$, is given by integrating the amplitudes at all frequencies over all frequencies.  When the weighting is included, that apparent total will be reduced below the value that would be obtained without the weighting.  It is important to note that a given average sound power level can correspond to different amplitudes at different frequencies, so when frequencies are relevant to the response, the $SPL_{av}$ is only partially relevant and may even be misleading.

In common usage today, the $SPL_{av}$, is measured by an instrument which does the integrating and filtering over all relevant frequencies and reports out one value over some prescribed sampling time.  This method of reporting is done for historical reasons since it has heretofore not been convenient, or perhaps necessary, to collect more detailed information.  In the case of wind turbines, amplitude as a function of frequency, and also amplitude as a function of time ("amplitude modulation"), may well be interesting and in

fact such information may now be readily obtained.  The authors should consider this possibility.

The types of wind turbine noise and the ways in which turbines produce sound are also relevant but were not discussed in the paper.  There are at least four types of noise: tonal, broadband, low-frequency, and impulsive.  The characteristics of the noise at a given location are a function of the turbine itself as well as the environment in which it operates.  For example, upwind vs. downwind rotors have different noise characteristics.  Higher blade tip speeds result in significantly higher noise production than slower tip speeds.  Rotors can be designed to operate with lower tip speeds but then the total blade area (and thus blade mass and so blade costs) will be higher, so there are trade-offs to be considered.  As has been widely reported, wind turbines may produce a thumping or whooshing sound; these have particular frequency and temporal features which are not apparent from the SPL_av data.  Wind shear and atmospheric stability may also affect the noise produced and how it is perceived.  The occurrence and characteristics of the sound may be weather dependent, and variations may occur in an apparently random and hence non-predictable manner.

The frequency characteristics of wind turbines, including low-frequency sound and infrasound, has been discussed by a variety of authors, but were not included in the paper- they should be.  Examples include the book Wind Turbine Noise by Bowdler and Leventhal (Multi-Science Publishing, 2011) and journal articles such as Leventhall, G. (2009). "Low Frequency Noise. What we know, what we do not know and what we would like to know." Journal of Low Frequency Noise, Vibration and Active Control, 28(2). 79-104 and Leventhall, G. (2006, June). "Infrasound from wind turbines - fact, fiction or deception." Canadian Acoustics, v 34, n 2, p 29-36.  Another report that has potentially useful references is the Wind Turbine Health Impact Study ([https://www.mass.gov/doc/wind-turbine-health-impact-study-report-of-independent-expert-panel/download](https://www.mass.gov/doc/wind-turbine-health-impact-study-report-of-independent-expert-panel/download)).

The use of predicted average sound pressure level may be less than ideal from a number of perspectives.  For regulators to use SPL_av as a basis of permitting may compel turbine designers and project developers to focus on reducing that average value.  If the frequency characteristics are actually more important, then the standards and regulations should be updated accordingly.  Turbine designers would have an additional impetus to fine-tune designs, perhaps to even consider "noise cancelling".  In summary, there may well be considerations other than purely technical regarding the wind turbine noise conundrum, but it should not be assumed that the average sound power level is the only relevant technical issue.  It is also not necessarily the case that more in-depth experimental that could be conducted, for example in the LaCour wind tunnel, would be pointless, as the authors seem to apply.